# Phytoconstituents Assisted Biofabrication of Copper Oxide Nanoparticles and Their Antiplasmodial, and Antilarval Efficacy: A Novel Approach for the Control of Parasites

**DOI:** 10.3390/molecules27238269

**Published:** 2022-11-27

**Authors:** Chidambaram Jayaseelan, Ahmed Abdulhaq, Chinnasamy Ragavendran, Syam Mohan

**Affiliations:** 1Department of Anatomy, Saveetha Medical College and Hospitals, Saveetha Institute of Medical and Technical Sciences (SIMATS), Chennai 602105, Tamil Nadu, India; 2Unit of Medical Microbiology, Department of Medical Lab Technology, Faculty of Applied Medical Sciences, Jazan University, Jazan 45142, Saudi Arabia; 3Department of Conservative Dentistry and Endodontics, Saveetha Dental College and Hospitals, Saveetha Institute of Medical and Technical Sciences (SIMATS), Chennai 600077, Tamil Nadu, India; 4Substance Abuse and Toxicology Research Centre, Jazan University, Jazan 45142, Saudi Arabia; 5School of Health Sciences, University of Petroleum and Energy Studies, Dehradun 248007, Uttarakhand, India; 6Center for Transdisciplinary Research, Department of Pharmacology, Saveetha Dental College, Saveetha Institute of Medical and Technical Science, Saveetha University, Chennai 600072, Tamil Nadu, India

**Keywords:** biofabrication, CuO NPs, antiplasmodial activity, *P. falciparum* (INDO), antilarval efficacy, therapeutic value

## Abstract

The present work aimed to biofabricate copper oxide nanoparticles (CuO NPs) using *Tinospora cordifolia* leaf extract. The biofabricated CuO NPs were treated against the malarial parasite of chloroquine-resistant *Plasmodium falciparum* (INDO) and the antilarval efficacy was evaluated against the malaria vector *Anopheles stephensi* and dengue vector *Aedes aegypti*. The prominence at 285 nm in the UV–visible spectrum helped to identify the produced CuO NPs. Based on the XRD patterns, the concentric rings correspond to reflections at 38.26° (111), 44.11° (200), 64.58° (220), and 77.34° (311). These separations are indicative of CuO’s face-centered cubic (fcc) structure. The synthesized CuO NPs have FTIR spectra with band intensities of 3427, 2925, 1629, 1387, 1096, and 600 cm^−1^. The absorbance band at 3427 cm^−1^ is known to be associated with the stretching O-H due to the alcoholic group. FTIR proved that the presence of the -OH group is responsible for reducing and capping agents in the synthesis of nanoparticles (NPs). The synthesized CuO NPs were found to be polymorphic (oval, elongated, and roughly spherical) in form with a size range of 11–47 nm and an average size of 16 nm when the morphology was examined using FESEM and HRTEM. The highest antiplasmodial efficacy against the chloroquine-resistant strain of *P. falciparum* (INDO) was found in the synthesized CuO NPs, with LC_50_ values of 19.82 µg/mL, whilst HEK293 cells are the least toxic, with a CC_50_ value of 265.85 µg/mL, leading to a selectivity index of 13.41. However, the antiplasmodial activity of *T. cordifolia* leaf extract (TCLE) and copper sulfate (CS) solution showed moderate activity, with LC_50_ values of 52.24 and 63.88 µg/mL, respectively. The green synthesized NPs demonstrated extremely high antilarval efficacy against the larvae of *An. stephensi* and *Ae. aegypti*, with LC_50_ values of 4.06 and 3.69 mg/L, respectively.

## 1. Introduction

Nanomaterials may be used in a wide variety of ways to enhance both the environment and human health. Due to their extensive variety of biological uses, copper and silver NPs are significantly more prominent among metal NPs [1]. They contain a variety of antibacterial effects that are well known, and the majority of them do not pose a threat to human health [2]. CuO NPs are of recent technological relevance and have gained increased attention because of their special characteristics. Additionally, compared to other NPs, CuO NPs were shown to be extremely sensitive to prokaryotes and eukaryotes. CuO NPs easily pass through biological barriers and into the intended organ [3]. Applications include being employed in paints, polymers, and fabrics as antimicrobial, antifouling, antibiotic, and antifungal agents [4]. CuO NPs are very important both theoretically and practically, and as a result, well-defined CuO nanostructures with a variety of morphologies have been produced. When NPs are created chemically, including the use of hazardous compounds, a number of issues arise. However, ecofriendly and economically advantageous nanoparticle manufacturing without the use of harmful chemicals is possible using green methods [5]. The creation of nanomaterials using greener approaches over physical and chemical techniques has a higher significance due to environmental friendliness, lower toxicity, lower cost, higher biocompatibility, and improved size-regulating characteristics [6]. Plants are among the ideal sources for the production of NPs. They also contain active phytocompounds that serve as reducing and capping agents, such as alkaloids, terpenoids, and steroids [7]. CuONPs were successfully synthesized through green process using *Enicostemma axillare* leaf extract [8], *Cardiospermum halicacabum* leaf extract [9], and *Morus alba* leaf extract [10]. CuONPs produced from the endophytic fungus *Aspergillus terreus* showed enhanced bioactivity, including antibacterial, antioxidant, and anticancer properties [11]. CuO NPs produced from *Ficus religiosa* leaf extract were investigated for anticancer efficacy using human A549 lung cancer cells by Sankar et al. [12]. Recently, the antibacterial potency of CuO NPs produced from *Tabernaemontana divaricata* leaf extract against a pathogen of the urinary tract was demonstrated [13].

Malaria is often regarded as among the most terrible tropical diseases at the global scale. Nearly half of the globe is at risk for malaria, which is endemic to areas of South and Central America, Africa, and South East Asia. Nearly one person dies from malaria every 30 s, mostly children and pregnant women [14]. Malaria is the third most widespread infectious illness among all age groups and the fourth most common cause of mortality in children under five [15]. Most of the cases and deaths occurred in 2019, with approximately 229 million cases and 409,000 deaths worldwide. The highest incidence was in Sub-Saharan Africa, where children under the age of five were most affected [16], and *Plasmodium* protozoan parasites were responsible. *Plasmodium malariae, P. ovale, P. vivax*, and *P. falciparum* are among the 100 species that have a prominent role in human disease manifestation. These are spread by bites from female *Anopheles* mosquitoes [17]. In Africa and India, *P. falciparum* is the most deadly and prevalent *Plasmodium* species, and it is the one that causes the majority of malaria-related mortality. Various *Plasmodium* species can be transferred to the host by a single mosquito bite [18].

Since they carry a variety of devastating human diseases, such as dengue fever, chikungunya, yellow fever, filariasis, Japanese encephalitis, and malaria, mosquitoes create a serious threat to public health [19]. In India, chikungunya, dengue, etc., affect approximately 1.5 million individuals annually [20]. Globally, 219 million cases have been documented, and the mortality rate is 4,350,000. Infectious disease caused by the dengue vector *Aedes aegypti* has increased by approximately 100 million in the past 30 years. Due to excessive and repeated application, several mosquito species have developed resistance to mosquito-controlling agents, such as chemically produced pesticides [21].

There is a renewed effort to develop chemicals derived from plants as they are considered more environmentally acceptable due to their inherent biodegradability and reduced toxicity to the majority of species. This is owing to the negative effects associated with synthetic pesticides, such as the development of pesticide-resistant strains, ecological imbalances, and harm to non-target organisms [22,23]. Because traditional medicine is more accessible and cost-effective than current pharmaceutical medicine, it is employed in rural regions. Therefore, traditional medicine is more user-friendly and has fewer adverse effects than contemporary pharmacological medicine [24]. Because of the limited availability and high cost of pharmaceutical drugs, an estimated two-thirds of the world’s population still use traditional medicinal treatments, primarily plants [25]. This explains why a lot of current research concentrates on natural chemicals and products produced from plants as they are readily available locally, easy to obtain, and may be chosen based on their ethnomedicinal benefits [26]. Because medicinal plants have a role in health care, scientific research into them has been initiated in many nations.

An essential medicinal herb *Tinospora cordifolia* Miers (Menispermaceae family) is used in the majority of Ayurvedic medicines [27,28]. Numerous therapeutic effects of this plant extract have been demonstrated, including general tonic, anti-inflammatory, antiarthritic, antimalarial, aphrodisiac [29], antiallergic [30], antidiabetic [31], and nephroprotective [32]. The aqueous extract of *T. cordifolia* has been said to have immunotherapeutic qualities; the active ingredient is purported to be a polysaccharide [33]. Berberine, tinosporin, tinosporal, tinosporaside, tinosporicacid, tinocordiofolioside, columbin, and other key chemical components of the plant all contribute to its medicinal properties [34].

In the present work, we describe the fabrication of CuO NPs by reducing the copper ions in the copper sulfate solution using an aqueous leaf extract from *T. cordifolia* that is cell free. It was shown that biologically produced NPs have strong larvicidal and antiplasmodial effects.

## 2. Results

### 2.1. UV–Vis Spectral Analysis and XRD Analysis

UV–Vis Spectra and XRD were used to characterize the biofabricated CuO NPs. Surface plasmon resonance (SPR) bands at 285 nm, which indicate metallic copper, were detected in UV–visible spectrum analysis, which provided preliminary confirmation of the production of CuO NPs (Figure 1). The patterns of the particles are shown in Figure 2 and the XRD findings show that they were crystalline in nature. The diffraction measurements were well matched with ICSD-087122 and point to a monoclinic structure. Based on the diffraction patterns, the concentric rings correspond to reflections at 38.26° (111), 44.11° (200), 64.58° (220), and 77.34° (311). These distinctions are consistent with CuO fcc structure.

### 2.2. FT-IR Analysis

Peaks were detected at 3427, 2925, 1629, 1387, 1096, and 600 cm^−1^ in areas within the range 500–4000 cm^−1^ of the FTIR spectrum in an attempt to identify the probable biomolecules responsible for the capping and efficient maintenance of the produced CuO NPs. (Figure 3).

### 2.3. SEM, TEM and EDX Analysis

FESEM has been employed to examine the morphology of biofabricated CuO NPs, and the micrograph indicates agglomerates with a variable size distribution. They were observed to have a smooth surface and a roughly spherical form (Figure 4). The EDX spectrum was captured at selected areas on the solid surface of purified CuO NPs to collect information on the atomic dispersion and elemental composition of the NPs. The EDX scan revealed a significant quantity of elemental copper signals and an oxygen peak on the surface of the produced CuO NPs (Figure 5). The size and form of CuO NPs were measured using the HRTEM investigation, and the photo shows that the particles are well dispersed and crystalline in nature. The HRTEM picture demonstrates that NPs are an interparticle distance apart instead of being in direct physical contact. CuO NPs had a variable morphology and were almost spherical in form, with an average diameter of 16 nm (Figure 6).

### 2.4. Antiplasmodial and Antilarval Efficacy of Synthesized Nanoparticles

The antiplasmodial effect of the biofabricated CuO NPs entirely inhibited the development of the chloroquine-resistant strain of *P. falciparum* (INDO) at a dose of 100 µg/mL, although the TCLE and CS solutions had the lowest activity (Figure 7). Cytotoxic effect of CuO NPs, TCLE, and CS solution exhibited poorer toxicity against HEK293 cells (Figure 8). The antimalarial and cytotoxicity potential of the synthesized CuO NPs TCLE, and CS solution were represented in Table 1. The highest antiplasmodial efficacy against the chloroquine-resistant strain of *P. falciparum* (INDO) was found in the synthesized CuO NPs, with an LC_50_ value of 19.82 µg/mL, whilst HEK293 cells are the least toxic, with a CC_50_ value of 265.85 µg/mL, leading to a selectivity index of 13.41. However, the antiplasmodial activity of TCLE and CS solution showed moderate effects, with IC_50_ values of 52.24 and 63.88 µg/mL, respectively. The cytotoxic effect of CuO NPs, TCLE, and CS solution showed less toxicity against HEK293 cells, with CC_50_ values of 265.85, 283.36, and 314.03 μg/mL, respectively.

High antilarval efficiency was found in the synthesized CuO NPs against the larvae of *An. stephensi* and *Ae. aegypti*, with LC_50_ values of 4.06 and 3.69 mg/L and LC_90_ values of 7.10 and 7.45 mg/L, respectively (Table 2). TCLE exhibited larvicidal properties against *An. stephensi* and *Ae. aegypti* and the results displayed a slight effect, with LC_50_ values of 54.98 and 59.63 mg/L and LC_90_ values of 105.86 and 114.974 mg/L, respectively. LC_50_ values of 74.79 and 77.81 mg/L and LC_90_ values of 125.16 and 138.15 mg/L were observed in the CS solution against the targeted larvae, respectively.

## 3. Discussion

The biological method of synthesizing NPs involves the use of extracts from various plant parts as a reducing agent of the metal ions. When producing metal NPs, biomolecules present in plant extracts serve as both reducing and stabilizing agents [35]. In this scenario, TCLE was used to synthesize bioactive CuO NPs, which were then tested utilizing a variety of methods. By using UV–visible spectrum analysis, which showed SPR bands at 285 nm, indicative of metallic copper, the synthesis of CuO NPs was preliminarily validated. The production and stability of metal NPs in aqueous solutions may be determined using UV–visible spectroscopy, which is a key process. Similar to this, the synthesized CuO NPs prepared from tea leaf extract had an absorption peak in the UV–visible spectrum at 271 nm [36]. It is clear from the findings of TCLE-synthesized CuO NPs that the concentric rings in the diffraction patterns correspond to reflections at 38.26° (111), 44.11° (200), 64.58° (220), and 77.34° (311). Based on the crystallinity of CuO NPs, XRD analysis revealed sharp peaks corresponding to (110), (111), (200), (202), (020), (202), (113), (311), (220), and (400) Bragg’s reflection. There are no additional diffraction peaks of other phases and all of the peaks may be indexed as the conventional monoclinic structure [37]. The crystal structure of the NPs was detected and verified using XRD methodology. In the present work, the FTIR spectrum was examined to find candidate biomolecules that could be in charge of effectively stabilizing and capping the CuO NPs synthesized by TCLE. Peaks at 3427, 2925, 1629, 1387, and 1096 cm^−1^ were detected. The pure CuO NPs exhibited an extensive band at approximately 3427 cm^−1^, which indicates the presence of the –OH group and the peak at 2916 cm^−1^ was attributed to a secondary amine. The strong band at 1629 cm^−1^ reveals the characteristic the C–O stretch. The peak at 1387 cm^−1^ indicates the C–N stretching mode of the aromatic amine group. The overall finding indicates the presence of certain proteins, terpenoids, or phenolic substances attached to the surface of CuO NPs. The free amino and carboxylic groups that have interacted with the copper surface may be the reason for the stability of CuO NPs. Additionally, the proteins in the medium help to stabilize the NPs by building a coat around them to avoid agglomeration [37]. The prominent diffraction pattern at 1610 cm^−1^ in the FTIR spectrum of copper NPs that were made using the latex of *Calotropis procera* is related to the binding of −NH−C=O to the metal NPs. Other notable FTIR bands include 1027 cm^−1^ (C-N stretching of amines), 2916 cm^−1^ (secondary amine), 1510 cm^−1^ (amide II), 1230 cm^−1^ (amide III), 1321 cm^−1^ (carboxylic acid), and 3423 cm^−1^ (alcohol), all of which strongly suggest the presence of protein on the nanoparticle surface [38]. FTIR spectroscopy confirmed the chemical constituents in the *Carica papaya* leaf extract minimize precursor and the production of CuO NPs. FTIR spectral peaks (1087 cm^−1^) demonstrate the existence of bands associated with amide N-H stretching (3444 cm^−1^), alkane C-H stretching (2926 cm^−1^), anhydride C=O bending (1880 cm^−1^), and C-O stretch [39]. In the FTIR spectrum of synthesized Cu_2_O NPs using *Tridax procumbens* leaf extract, the absorption peaks were located mainly at 3444, 1644, 1337 and 618 cm^−1^ in the region. The prominent spectrum of hydrogen-bonded OH groups found in the aqueous phase is the peak at 3444 cm^−1^. The presence of (-COO-) carboxylate ions, which stabilize the CuO NPs, was confirmed by the existence of the peaks at 1644 cm^−1^ (asymmetric) and 1337 cm^−1^ (symmetric) [40]. According to Sankar et al. [12], the produced CuO NPs from *Ficus religiosa* leaf extract was spherical in form and uniformly dispersed throughout the colloidal solution. EDX study found the large proportion of elemental copper and oxide peaks, and the nanoparticle form significantly altered their optical and electrical features. CuO NPs were produced from *Acalypha indica* leaf extract, and SEM analysis indicated that the particles are well spread, spherical, and range in size from 26 to 30 nm. The EDX spectrum demonstrates a significant copper signal and proves the production of CuO NPs [41]. The TEM images of *T. divaricata* leaf extract synthesized CuO NPs reveal that the materials were of nearly spherical in shape with a size of 48 ± 4 nm, which was in good agreement with the XRD particle size [13]. The NPs appear to have formed relatively wide, quasi-linear superstructures instead of a compact, densely packed assembly [42]. The biological production of CuO NPs using an aqueous extract of *A. indica* leaf was reported by Sivaraj et al. [41]. The TEM investigation revealed that the synthesized particles were extremely stable, spherical, and between 26 and 30 nm in size. With the use of TEM, the shape and size of the CuO-NPs are produced from the fruit and leaf of *Rubus glaucus*. The majority of the NPs displayed on the micrograph are spherical, well organized, and have mean sizes of 45 and 53 nm, respectively. The partially crystalline properties of the produced CuO NPs were confirmed by the presence of the SAED pattern [43].

The development of drug resistance is the primary drawback of conventional malaria treatment. The severity of malaria has worsened in many endemic regions of the world as a result of drug resistance to quinine, chloroquine, primaquine, and mefloquine [44]. Antimalarial drugs are less effective due to certain aspects including limited bioavailability, fast metabolism, and poor absorption [45,46]. However, NPs are excellent because of their distinct and spectacular properties, which include tiny size, good bioavailability, lower toxicity, avoidance of drug resistance, and site-specific drug delivery [47]. In our investigation, it was observed that the TCLE-synthesized CuO NPs exhibited greater antiplasmodial activity against *P. falciparum* compared to TCLE. Because the plant phytoconstituents were coated, NPs were produced that could provide a wide range of distinct microenvironments and change the physicochemical characteristics, resulting in increased anticancer activity. CuO NP treatments, meanwhile, worked less well than chloroquine. The antimalarial activity of an expanding variety of nanomaterials against *P. falciparum* has recently been explored. While plant extracts or amylase alone did not exhibit any action up to 40 mg/mL, AgNPs synthesized utilizing leaf extracts of ashoka, neem, and alpha amylase inhibit the development of *P. falciparum*, with IC_50_ values of 3.75 (amylase NP), 8 (ashoka NP), and 30 µg/mL, respectively [48].

Presently, *An. stephensi* and *Ae. aegypti* 4th instar larvae were treated with CuO NPs, TCLE, and CS solution. When compared to TCLE and CS solution, CuO NPs exhibited the highest level of larvae mortality. Similarly, the maximum mortality was exhibited in the *Nelumbo nucifera* leaf aqueous extract and produced silver NPs (Ag NPs) against the larvae of *An. subpictus* (LC_50_= 11.82, and 0.69 ppm) and against the larvae of *Cx. quinquefasciatus* (LC_50_ = 13.65, and 1.10 ppm) [49]. In our previous investigations, we have reported that the antilarval effect of *T. cordifolia* extract synthesized Ag NPs against the larvae of *An. subpictus* and *Cx. quinquefasciatus*, with LC_50_ values of 6.43 and 6.96 mg/L [50]. NPs easily penetrate the bodies of invertebrates through the exoskeleton. Once they have entered into the insect cell, the NPs interact with molecules such as DNA and proteins to alter their structure and therefore function [51]. It should be emphasized that plant-based NPs are very harmful to different mosquito larvae but have little impact on non-target species such as fish and other aquatic arthropods [52,53].

## 4. Materials and Methods

### 4.1. Collection and Preparation of Plant Extract

The fresh leaves of *Tinospora cordifolia* were collected from the Christian Medical College, Vellore, India. To clean the dust, the leaves were thoroughly washed in tap water for 10 min, then rinsed in deionized water. In a 250 mL Erlenmeyer flask, 10 g of finely chopped *T. cordifolia* leaves was combined with 100 mL of deionized water to produce an aqueous *T. cordifolia* leaf extract (TCLE). The mixture was then boiled at 60 °C for 30 min, cooled at 25 ± 2 °C and filtered using No.1 Whatman filter paper. The filtrate was stored at 2–8 °C for further analysis.

### 4.2. Synthesis of CuO NPs

For the production of CuO NPs, an Erlenmeyer flask containing 15 mL of TCLE was added to 85 mL of 5 mM copper sulfate (Sigma-Aldrich, St.Louis, MO, USA) solution that had been rapidly agitated. The mixture was then heated at 60 °C for 30 min. The blue-colored copper sulfate solution turned into green color, which indicates the formation of copper hydroxide. This possibly occurred due to the reaction of CuSO_4_.5H_2_O with the hydroxyl anion (OH^−^) generated in aqueous solution, forming copper hydroxide (Cu(OH)_2_). Then, the extracted phytochemicals (oligosaccharides, amino acids, phenols, and flavonoids) acted as oxidizing agents, reducing Cu(OH)_2_ to CuO NPs and forming a colloidal solution [54]. The resulting solution was examined on a regular basis using UV–visible spectroscopy in the 200–700 nm wavelengths. CuSO_4_ in aqueous solution (5 mM) served as the control. The synthesized CuO NPs were centrifuged at 5000× *g* for 15 min to prepare the pellet, and the residual biomass was washed off with deionized water. The pellet was obtained after drying the viscous layer in the oven at 50 °C for 24 h [8].

### 4.3. Characterization of CuO NPs

The UV absorbance spectra of fabricated CuO NPs were analyzed using a Schimadzu 1601 spectrophotometer with a 1 nm resolution. The produced NPs were characterized by XRD analysis. The spectra were captured using a Philips^®^ PW 1830 X-ray generator and a Bruker AXSD8 Advance X-ray diffractometer. The synthesized CuO NPs were characterized using FTIR analysis with a 350–4000 cm^−1^ scanning range and a 4 cm^−1^ resolution. The diffuse reflectance mode and a resolution of 4 cm^−1^ in KBr pellets were used for these observations using a Perkin-Elmer Spectrum One instrument. Using FESEM and a JEOL JSM 6700F, the surface morphology of NPs was investigated (JEOL, Tokyo, Japan). The chemical build and quality of the produced NPs were examined using EDX (a German-made RONTEC EDX equipment, Model QuanTax 200). The product shape and crystal structure were evaluated using transmission electron microscopy (TEM). The device was a JEOL 2000Fx-II analytical TEM with a W-source and a point-to-point resolution of 2, operating at 200 kV.

### 4.4. In Vitro Cultivation of Plasmodium Falciparum

The chloroquine (CQ)-resistant strain of *Plasmodium falciparum* (INDO) was obtained from the National Institute of Malaria Research (NIMR), Delhi, India. *P. falciparum* (INDO) strain was cultivated in a continuous culture using the method of Trager and Jensen [55], with a few tiny adjustments. Fresh O+ve group human erythrocytes dissolved at 4% hematocrit were used to maintain the cultures. The complete medium used for the cultures was 16.2 g/L RPMI 1640 containing 25 mM HEPES, 11.11 mM glucose (Gibco), 0.2% sodium bicarbonate (Sigma), 0.5% (*w/v*) Albumax I (Gibco), 45 g/mL hypoxanthine (Sigma), and to propagate the culture, the used medium was changed each day with brand-new, whole medium. Giemsa stained blood samples undergo microscopic screening to track parasitemia. Treatment with 5% sorbitol resulted in the parasite achieving the synchronized circle stage [56].

### 4.5. Drug Dilutions

Stock solutions of CuO NPs, TCLE, and CS solution were prepared in milli-Q water. All stocks were then diluted with the culture medium to achieve the required drug concentrations [48].

### 4.6. Assay for Antiplasmodial Activity

SYBR Green I-based fluorescence assessment was evaluated for antiplasmodial screening, as earlier described by Smilkstein et al. [57]. To make the stock solution, 25 mg of each NGP was individually suspended in 1 mL of deionized water and thoroughly combined using a vortex mixer. Twofold aliquots of the stock sample were produced in cRPMI. Using a multi-pipette, 4 µL of NPs in triplicates at different doses was injected into each well of the 96-well plate. As positive (0% growth) and negative controls (100% growth), respectively, CQ at 1 M and 0.4% DMSO (*v/v*) were used. A volume of 96 µL of sorbitol-synchronized ring stage parasites (2% hematocrit and 1% parasitemia) was supplied to each well after the addition of each sample group, and all the wells were then cultured for 48 h at 37 °C in an environment of 5% O_2_, 5% CO_2_, and 90 % N_2_. Following the growth period, 100 µL of SYBR green I buffer [0.2 L of 10,000 SYBR Green I (Invitrogen) per mL of lysis buffer] was added to each well, mixed thoroughly, and incubated at 37 °C in the dark for one hour. The lysis buffer enclosed Tris (20 mM; pH 7.5), EDTA (5 mM), saponin (0.008%; *w/v*), and Triton X. With the excitation and emission wavelengths centered at 485 and 535 nm, respectively, a Victor fluorescence multi-well plate scanner (PerkinElmer, Waltham, MA, USA) was used to detect fluorescence. The numbers in each well were subtracted from the fluorescence counts for CQ. IC_50_ (the 50% inhibitory percentage) data were estimated using the IC Estimator-version 1.2 software (http://www.antimalarial-icestimator.net/MethodIntro.htm) (Free Software Foundation, Boston, MA, USA) by plotting the fluorescence counts against the drug content.

### 4.7. Cytotoxicity of NPs

Human embryonic kidney (HEK 293) cells were maintained in DMEM holding 10% fetal bovine serum, 0.21% sodium bicarbonate (Sigma), and 50 μg/mL gentamicin to determine the cytotoxic activity of the produced NPs on host cells. In a nutshell, 96-well flat-bottomed tissue culture dishes were seeded with cells placed in DMEM (10^4^ cells/100 μL/well). After 24 h, the medium was swapped with 96 µL of new medium, and 4 µL of the escalating test sample levels was added, and the plate was maintained for another 24 h at 37 °C with 5% CO_2_. Controls included 10% DMSO (zero growth) and 100% DMEM (100% growth). Each well received 20 microliters of a stock MTT solution (5 mg/mL in phosphate buffered saline), which was then gently mixed and left to sit for an additional 4 h. The upper layer was removed after spinning the plate for five minutes at 1500 rpm, and 200 μL of 100% DMSO (a stopping agent) was then added. A microtiter plate scanner (Versa max tunable multi-well plate reader) was employed to quantify the development of formazan at 570 nm. The examination of dose–response graphs was used to establish the drug’s 50% cytotoxic concentration (CC_50_). The ratio of the CC_50_ HEK293/IC_50_ *Pf*INDO was used to estimate the selectivity index [55].

### 4.8. Vectors Rearing

Larvae of *Anopheles stephensi* and *Aedes aegypti* were collected from a rice field to a stagnant water site in Gudiyatham and confirmed at the Zonal Entomological Research Centre in Vellore, Tamil Nadu, to develop the colony. The larvae were cultured in soft and enamel trays with tap water. They were reared for and developed in the laboratory using the methodology described by Kamaraj et al. [58].

### 4.9. Larvicidal Bioassay

The WHO procedure was used to evaluate the larvicidal activity [59]. Each test sample (CuO NPs, TCLE, and CS) was individually dissolved in 100 mL of distilled water to prepare the stock solution. Using double-distilled water as a solvent, the appropriate doses were obtained from the stock solution. Twenty mosquito larvae were used in the antilarval experiment, along with different test sample concentrations, in 200 mL of sterilized double-distilled water, which was then placed in an ambient chamber set at 25 °C with a 16:8 h light/dark cycle. For each test, there were five repetitions of each dose of distilled water in a set of control groups. *Ae. aegypti* and *An. stephensi* 4th instar larval mortality was tested after 24 h to assess the acute toxicities, and the death of the larvae was confirmed by stoppage of movement when touched by a needle.

### 4.10. Dose–Response Bioassay

CuO NPs, TCLE, and CS were put through a dose–response experiment for larvicidal efficacy against the larvae of *Ae. aegypti* and *An. stephensi* based on the preliminary screening results. For larvicidal activities, various concentrations ranging from 30 to 150 mg/L for TCLE and CS and 2.0 to 10 mg/L for CuO NPs, were prepared. After 24 h, the number of dead larvae was calculated, and the mortality rate was plotted using the median of five replicates. Using the SPSS 2007 statistical analysis program, the mean larval mortality data were exposed to probit analysis in order to determine the LC_50_, LC_90_, and 95% fiducial limits of upper and lower confidence levels.

## 5. Conclusions

In the present study, the biofabrication of TCLE-mediated CuO NPs was achieved. The UV–visible spectral findings displayed the optical density band at 285 nm and the XRD confirms the crystalline structure of the CuO particles. The TEM investigation showed that the morphology of nanomaterials was irregular and nearly spherical in shape, with a mean of 16 nm. The produced NPs exhibited strong antiplasmodial effects against the chloroquine-resistant strain of *P. falciparum* (INDO) and also exhibited insignificant toxicity against HEK293 cells, leading to a high selectivity index. On the other hand, CuO NPs showed better larvicidal mortality against *An. stephensi* and *Ae. aegypti* than the plant extract. As an outcome of these findings, *T. cordifolia* aqueous leaf extract might be employed as an excellent reducing agent in the synthesis of NPs. To assess the chances of producing a product and its economic viability for parasite control using ecofriendly methods, more research is required.

## Figures and Tables

**Figure 1 molecules-27-08269-f001:**
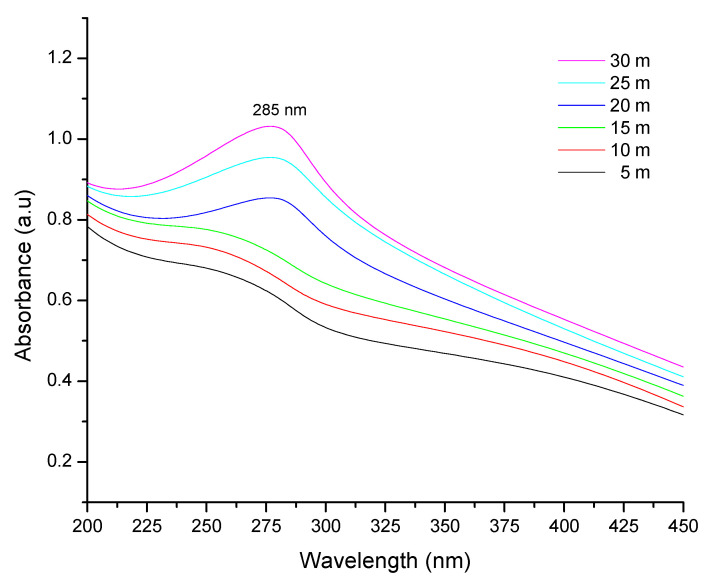
UV–visible absorption spectra of synthesized CuO NPs.

**Figure 2 molecules-27-08269-f002:**
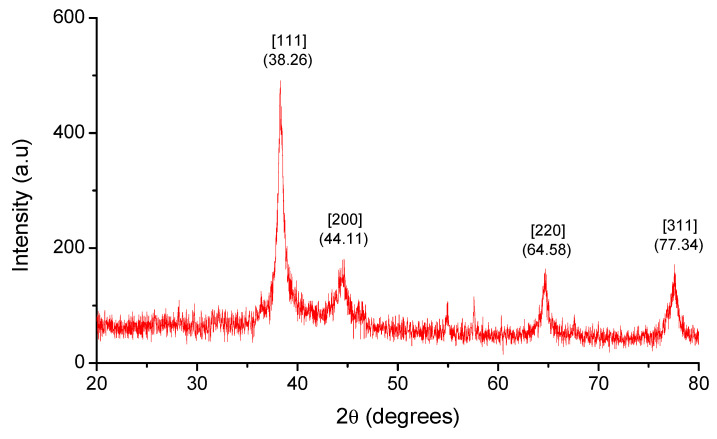
XRD spectra of synthesized CuO NPs.

**Figure 3 molecules-27-08269-f003:**
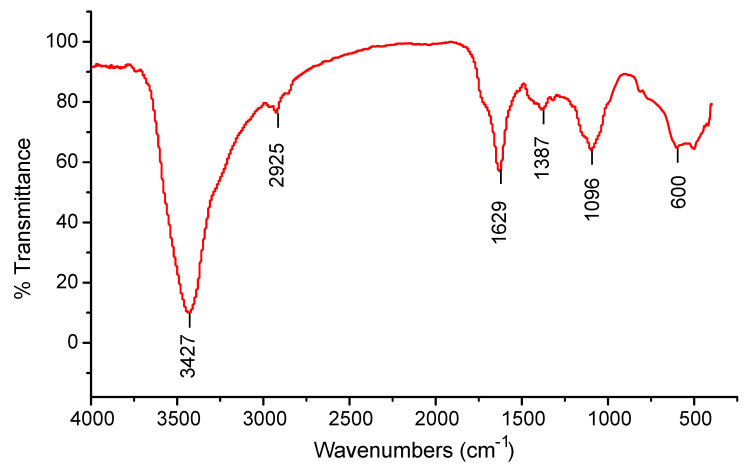
FTIR spectra of synthesized CuO NPs.

**Figure 4 molecules-27-08269-f004:**
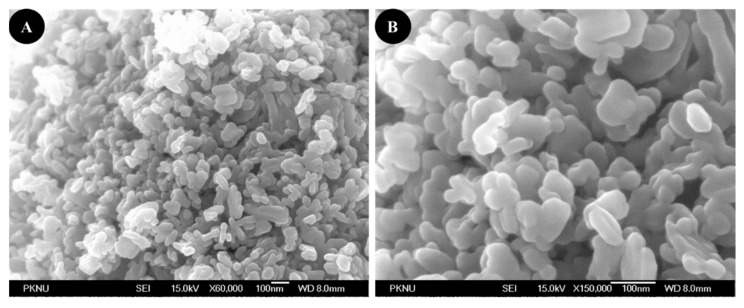
FESEM images of synthesized CuO NPs at (**A**) 60,000× and (**B**) 150,000× magnifications.

**Figure 5 molecules-27-08269-f005:**
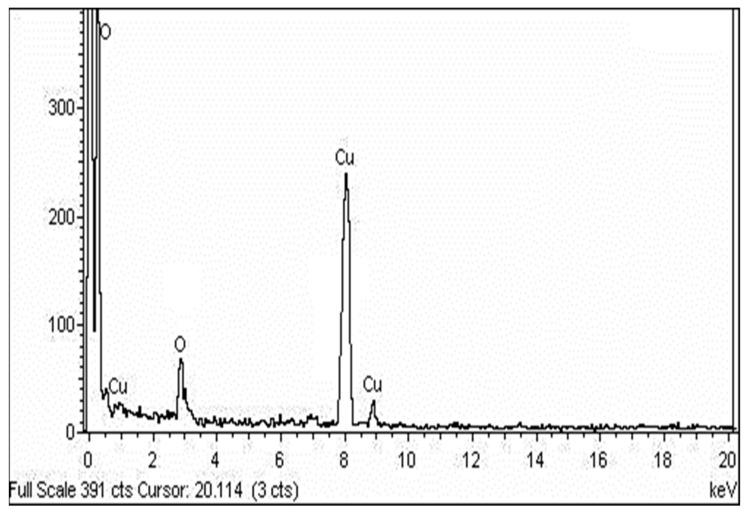
EDX spectra of synthesized CuO NPs.

**Figure 6 molecules-27-08269-f006:**
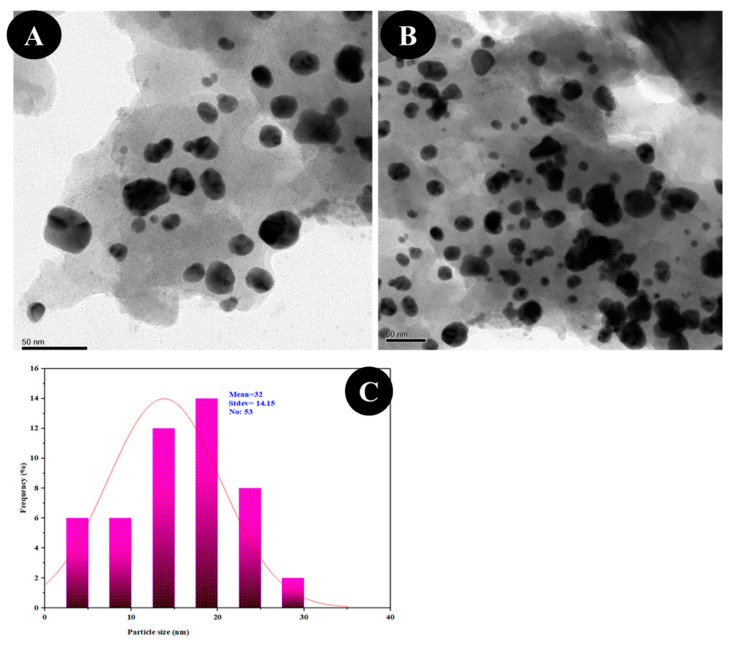
The TEM images of CuO NPs (**A**,**B**) showing morphology and (**C**) particle size distribution.

**Figure 7 molecules-27-08269-f007:**
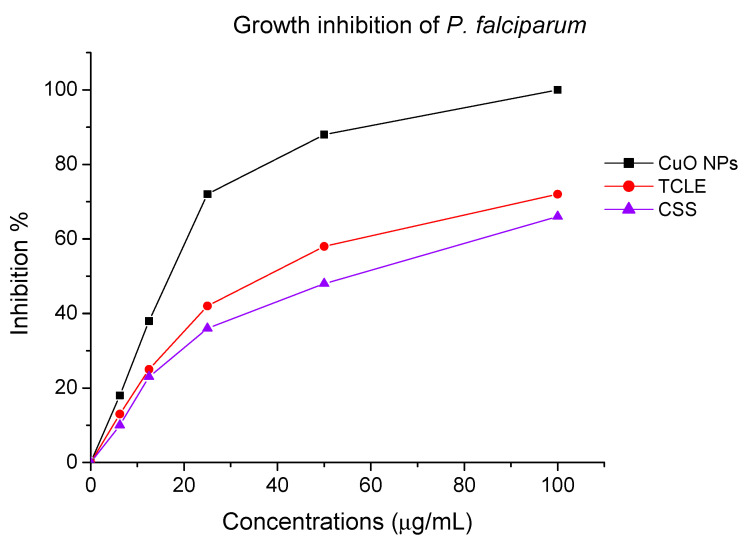
Antiplasmodial effect of biofabricated copper oxide nanoparticles (CuO NPs), *T. cordifolia* leaf extract (TCLE), and copper sulfate (CS) solution.

**Figure 8 molecules-27-08269-f008:**
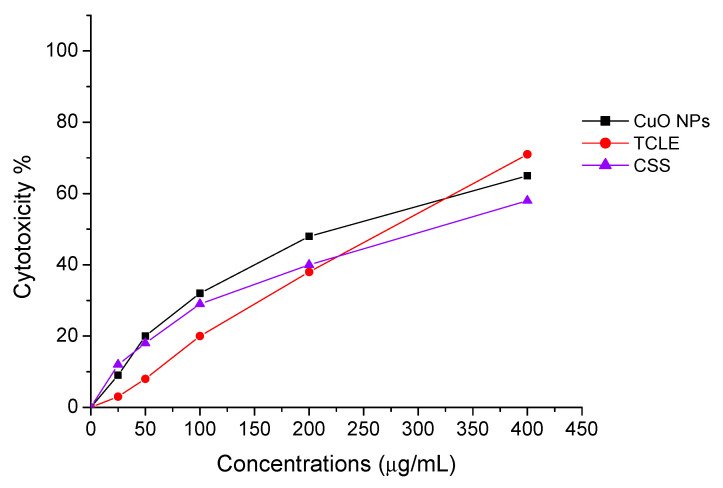
Cytotoxic effect of biofabricated copper oxide nanoparticles (CuO NPs), *T. cordifolia* leaf extract (TCLE), and copper sulfate (CS) solution.

**Table 1 molecules-27-08269-t001:** Antiplasmodial and cytotoxic effect of biofabricated copper oxide nanoparticles (CuO NPs), *T. cordifolia* leaf extract (TCLE), and copper sulfate (CS) solution.

Sample	Antiplasmodial Activity IC_50_ (µg/mL)	Cytotoxicity CC_50_ (µg/mL)	Selectivity Index
	*pf*INDO	HEK293	(CC_50_ HEK293/IC_50_ *Pf*INDO)
CuO NPs	19.82	265.85	13.41
TCLE	52.24	283.36	5.42
CS	63.88	314.03	4.91

**Table 2 molecules-27-08269-t002:** Antilarval efficacy of biofabricated copper oxide nanoparticles (CuO NPs), *T. cordifolia* leaf extract (TCLE), and copper sulfate (CS) solution against *An. stephensi* and *Ae. Aegypti*.

Species	Sample	LC_50_ (mg/L)	95% Confidence Limit		LC_90_ (mg/L)	95% Confidence Limit		r^2^	χ^2^ *d.f.* = 4
			Lower	Upper		Lower	Upper		
*An. stephensi*	CuO NPs	4.06	3.69	4.40	7.10	6.63	7.70	0.945	1.940
	TCLE	54.98	48.46	60.69	105.86	98.27	115.73	0.952	2.560
	CS	74.79	60.67	87.61	125.16	108.79	155.10	0.987	6.386
*Ae. aegypti*	CuO NPs	3.69	2.25	4.66	7.45	6.27	9.87	0.973	6.849
	TCLE	59.63	52.94	65.55	114.97	106.79	125.63	0.980	3.853
	CS	77.81	71.53	83.77	138.15	128.81	150.32	0.994	0.97

No mortality was observed in the control. LC_50_, lethal concentration that kills 50% of the exposed organisms; LC_90_, lethal concentration that kills 90% of the exposed organisms; r^2^, regression coefficient; χ^2^, Chi square value; *d.f.*, degrees of freedom.

## Data Availability

The datasets used and/or analyzed during the current study are available from the corresponding author on reasonable request.

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
