# Peer review of "Phytoconstituents Assisted Biofabrication of Copper Oxide Nanoparticles and Their Antiplasmodial, and Antilarval Efficacy: A Novel Approach for the Control of Parasites"

_molecules, 2022, doi:10.3390/molecules27238269_

Round 1
Reviewer 1 Report
The article describes the phytoextract-assisted synthesis of CuO NPs and their antiplasmodial and antilarval applications. The article abridged useful information but some parts of this draft should be revised. That is;
1. In Abstract:
- The author claimed that "FTIR proved that the presence of -OH group is responsible for capping and stabilizing agents for the synthesis of nanoparticles”, however, there are no studies that demonstrate stability data in this article. Reword the sentence.
- Synthesized CuO NPs are polymorphic (oval, elongated, roughly spherical), not spherical, as can be seen from SEM/TEM photos.
- It is better to mention the range of CuO NP’s range size along with the average size.
2. In Introduction:
- Line no. 58-61, add relevant references for these statements.
- Line no. 63, 75-76, and 77: Scientific names should be in italics.
- Describe shortly the eco-friendly synthesis of CuO NPs using various plant extracts, and their biological applications, specifically on the cytotoxicity and control of parasites with relevant references.
- Tinospora cordifolia's ethnomedicinal advantages or significance need to be explained briefly with appropriate references.
3. In Material and Methods:
- Line no. 335: " then quickly dried in deionized water”, correct it.
- Line no. 335-338: rephrase statements in a sequence of steps followed for extract preparation, such as plant material collection, washing, drying, chopping, aqueous mixture preparation, and heating/boiling mixture for final extract preparation, etc.
- In 4.2. Synthesis of CuO NPS: Chemicals and instrument details like manufacturer, make model, etc. are missing.
- In 4.3. Characterization of CuO NPs: What procedures were applied for preparing a dried sample of CuO NPs? Make and model of Instrument? FTIR measurement unit needs to correct (line no. 357).
- Sample preparation methods and procedures for each test should describe or provide references with the appropriate citations.
- Source (procured or obtained from) of Plasmodium falciparum?
4. In Results:
- Line no.: “Many instruments........... CuO NPs", this sentence does not make sense, better to replace it with a focus on UV-vis and XRD instruments.
- Line no. 111-112: " The ideal conditions required to produce CuO NPs were 60°C, pH6, 5 mM CuSO4, and a 30 min incubation time.”, is this synthesis method procedure or result? Rephrase the sentence.
- If possible authors can add FTIR spectra of Plant (Tinospora cordifolia) extract.
5. In Discussion:
- Line no. 255-279: correct FTIR measurement unit.
- Line no. 281: Scientific name eg. Ficus religiosa should be in italics.
- Line no. 279-280 and 292: Use et al. instead of listing all author’s names of reference articles when citing in text. Follow journal guideline for citation in text and reference.
- Line no. 299-300: Rewrite sentence or provide reference for the statement.
- Line no. 300-301: What are the antimicrobial drug, list few names with reference, and Where it has less effect, on microbes or parasites? Antimicrobial drugs or antiplasmodial drugs? There is a missing connection between your findings and provided references.
Author Response
Dear Reviewer the reply is attached

Reviewer 2 Report
Comments to the Author
This article describes the bio-fabrication of copper oxide nanoparticle synthesis using T. cordifolia leaf. The following comments should be addressed before publications
1) Novelty of the manuscript must be better emphasized
2) The importance of this study should be more clearly explained in the introduction section
3) Justify the selection of T. cordifolia leaves in the introduction.
4) Brief reason why was extract used quickly after preparation, within 1h. line 338
5) How can the mixture be boiled at 60 °C? -line 337
6) Synthesis part is not clear add some more details such as the UV-visible spectrum
7) Mention the medium used to dissolve nanoparticles for biological applications
8) Add some more discussion points to the materials characterization part
9) Typological errors should be revised neatly
10) Update references
Author Response
Dear Reviewer, please find the attached file

Round 2
Reviewer 2 Report
All queries are answered.